# Quantifying transmission dynamics of acute hepatitis C virus infections in a heterogeneous population using sequence data

**Gonché Danesh**[1]*, **Victor Virlogeux**[2], **Christophe Ramière**[3], **Caroline Charre**[3], **Laurent Cotte**[4☯], **Samuel Alizon**[1☯]

1 MIVEGEC, CNRS, IRD, Université de Montpellier – Montpellier, France, 2 Clinical Research Center, Croix-Rousse Hospital, Hospices Civils de Lyon – Lyon, France, 3 Virology Laboratory, Croix-Rousse Hospital, Hospices Civils de Lyon – Lyon, France, 4 Infectious Diseases Department, Croix-Rousse Hospital, Hospices Civils de Lyon – Lyon, France

☯ These authors contributed equally to this work.
* gonche.danesh@gmail.com

## Abstract

Opioid substitution and syringes exchange programs have drastically reduced hepatitis C virus (HCV) spread in France but HCV sexual transmission in men having sex with men (MSM) has recently arisen as a significant public health concern. The fact that the virus is transmitting in a heterogeneous population, with different transmission routes, makes prevalence and incidence rates poorly informative. However, additional insights can be gained by analyzing virus phylogenies inferred from dated genetic sequence data. By combining a phylodynamics approach based on Approximate Bayesian Computation (ABC) and an original transmission model, we estimate key epidemiological parameters of an ongoing HCV epidemic among MSMs in Lyon (France). We show that this new epidemic is largely independent of the previously observed non-MSM HCV epidemics and that its doubling time is ten times lower (0.44 years *versus* 4.37 years). These results have practical implications for HCV control and illustrate the additional information provided by virus genomics in public health.

## Author summary

Lyon (France) is witnessing a new epidemic of hepatitis C virus infection, which appears to be fuelled by sexual transmission. Upon detection, patients are found to belong to two main risk groups. The first group is referred to as non-MSM and typically corresponds to HIV-negative patients infected through nosocomial transmission or with a history of opioid intravenous drug use or blood transfusion or patients with haemophilia. The second group is more recent and mainly corresponds to Men Having Sex with Men (MSM) who are HIV-infected or HIV-negative MSMs. They tend to be detected during or shortly after the acute HCV infection phase and to use recreational drugs such as cocaine or cathinones. By taking advantage of recent developments in the emerging field of phylodynamics, we combine this patient information with virus sequence data to estimate key

**Data Availability Statement:** Scripts and data are available at https://zenodo.org/record/4314714#. X9O3RYZCdhF.

**Funding:** This reasearch was funded by the Fondation pour la Recherche Médicale (FRM grant number ECO20170637560) to GD. The funders had no role in study design, data collection and analysis, decision to publish, or preparation of the manuscript.

**Competing interests:** The authors have declared that no competing interests exist.

properties of the epidemics. We show that the current HCV spread via sexual transmission and MSM hosts is comparable to that before the advent of third-generation detection tests. We also find that the duration of the effective infectious period in MSM hosts is comparable to that of the acute phase. These results have timely public health implications, one of which is that treatment upon detection is necessary to slow down the ongoing HCV epidemics in Lyon.

## Background

It is estimated that 71 million people worldwide suffer from chronic hepatitis C virus (HCV) infections [1, 2]. The World Health Organisation (WHO) and several countries have issued recommendations towards the 'elimination' of this virus, which they define as an 80% reduction in new chronic infections and a 65% decline in liver mortality by 2030 [2]. HIV-HCV coinfected patients are targeted with priority because of the shared transmission routes between the two viruses [3] and because of the increased virulence of HCV in coinfections [4–6]. Successful harm reduction interventions, such as needle-syringe exchange and opiate substitution programs, as well as a high level of enrolment into care programs for HIV-infected patients, have led to a drastic drop in the prevalence of active HCV infections in HIV-HCV coinfected patients in several European countries during the recent years [7–10]. Unfortunately, this elimination goal is challenged by the emergence of HCV sexual transmission, especially among men having sex with men (MSM). This trend is reported to be driven by unprotected sex, drug use in the context of sex ('chemsex'), and potentially traumatic practices such as fisting [11–13]. The epidemiology of HCV infection in the Dat'AIDS cohort has been extensively described from 2000 to 2016 [14–16]. The incidence of acute HCV infection has been estimated among HIV-infected MSM between 2012 and 2016, among HIV-negative MSM enrolled in PrEP between in 2016–2017 [13] and among HIV-infected and HIV-negative MSMs from 2014 to 2017 [17]. In the area of Lyon (France), HCV incidence has been shown to increase concomitantly with a shift in the profile of infected hosts [17]. The incidence of first HCV infection regularly increased in HIV-positive MSM from the area of Lyon [13]. Understanding and quantifying this recent increase is the main goal of this study.

Several modelling studies have highlighted the difficulty to control the spread of HCV infections in HIV-infected MSMs in the absence of harm reduction interventions [12, 18]. Furthermore, we recently described the spread of HCV from HIV-infected to HIV-negative MSMs, using HIV pre-exposure prophylaxis (PrEP) or not, through shared high-risk practices [17]. More generally, an alarming incidence of acute HCV infections in both HIV-infected and PrEP-using MSMs was reported in France in 2016–2017 [13]. Additionally, while PrEP-using MSMs are regularly screened for HCV, those who are HIV-negative and do not use PrEP may remain undiagnosed and untreated for years. In general, we know little about the population size and practices of HIV-negative MSM who do not use PrEP. All these epidemiological events could jeopardize the goal of HCV elimination by creating a large pool of infected and undiagnosed patients, which could fuel new infections in intersecting populations. Furthermore, the epidemiological dynamics of HCV infection have mostly been studied in intravenous drug users (IDU) [19–22] and the general population [23, 24]. Results from these populations are not easily transferable to other populations, which calls for a better understanding of the epidemiological characteristics of HCV sexual transmission in MSM.

Given the lack of knowledge about the focal population driving the increase in HCV incidence, we analyse virus sequence data with phylodynamics methods. This research field has

been blooming over the last decade and hypothesizes that the way rapidly evolving viruses spread leaves 'footprints' in their genomes [25–27]. By combining mathematical modelling, statistical analyses and phylogenies of infections, where each leaf corresponds to the virus sequence isolated from a patient, current methods can infer key parameters of viral epidemics. This framework has been successfully applied to other HCV epidemics [28–31], but the ongoing one in Lyon is challenging to analyze because the focal population is heterogeneous, with non-MSM hosts, which are typically HIV-negative patients infected through nosocomial transmission or with a history of opioid intravenous drug use or blood transfusion or patients with hemophilia, and MSM hosts. For these MSM hosts, transmission appears to take place during sexual contact but host profiles established by field epidemiologists based on interviews and risk factors yield a less clearcut picture. These MSM hosts include both HIV-infected and HIV-negative MSM, detected during or shortly after acute HCV infection phase, potentially using recreational drugs such as cocaine or cathinones in the context of 'chemsex'. Our phylodynamics analysis relies on an Approximate Bayesian Computation (ABC, [32]) framework that was recently developed and validated using a simple Susceptible-Infected-Recovered (SIR) model [33].

Assuming an epidemiological transmission model with two host types, non-MSM and MSM (see the Material and methods), we use dated virus sequences to estimate the date of onset of the HCV epidemics in non-MSM and MSM hosts, the level of mixing between hosts types, and, for each host type, the duration of the infectious period and the effective reproduction ratio (i.e. the number of secondary infections, [34]). To validate our results we performed a parametric bootstrap analysis, we tested the sensitivity of the method to differences in sampling proportions between the two types of hosts. We also tested the sensitivity of the method to phylogenetic reconstruction uncertainty, and we performed a cross-validation analysis to explore the robustness of our inference framework. We find that the doubling time of the epidemics is one order of magnitude lower in MSM than in non-MSM hosts, therefore emphasising the urgent need for public health action.

## Results

The phylogeny inferred from the dated virus sequences shows that MSM hosts (in red) tend to be grouped in clades (Fig 1). This pattern suggests a high degree of assortativity in the epidemics (i.e. hosts tends to infect hosts from the same type). The ABC phylodynamics approach allows us to go beyond a visual description and to quantify several epidemiological parameters.

As for any Bayesian inference method, we need to assume a prior distribution for each parameter. These priors, shown in grey in Fig 2, are voluntarily designed to be large and uniformly distributed to be as little informative as possible. One exception is the date of onset of the epidemics, for which we use the output of the phylogenetic analysis conducted in Beast (see the Material and methods) as a prior. We also assume the date of the MSM hosts epidemics to be after 1997 based on epidemiological data.

The inference method converges towards posterior distributions for each parameter, which are shown in red in Fig 2 and summarized in Tables 1 and 2. The estimate for the origin of the epidemic in non-MSM hosts is $t_0$ = 1957.47 [1948.61; 1961.96] (numbers in brackets indicate the 95% Highest Posterior Density, or HPD). For the MSM host type, we were not able to estimate when the epidemic ($t_2$) has started.

We find the level of assortativity between host types to be high for non-MSM ($a_1$ = 0.94 [0.83; 1.0]) as well as for MSM hosts ($a_2$ = 0.92 [0.81; 0.99]). Therefore, hosts mainly infect hosts from the same type.

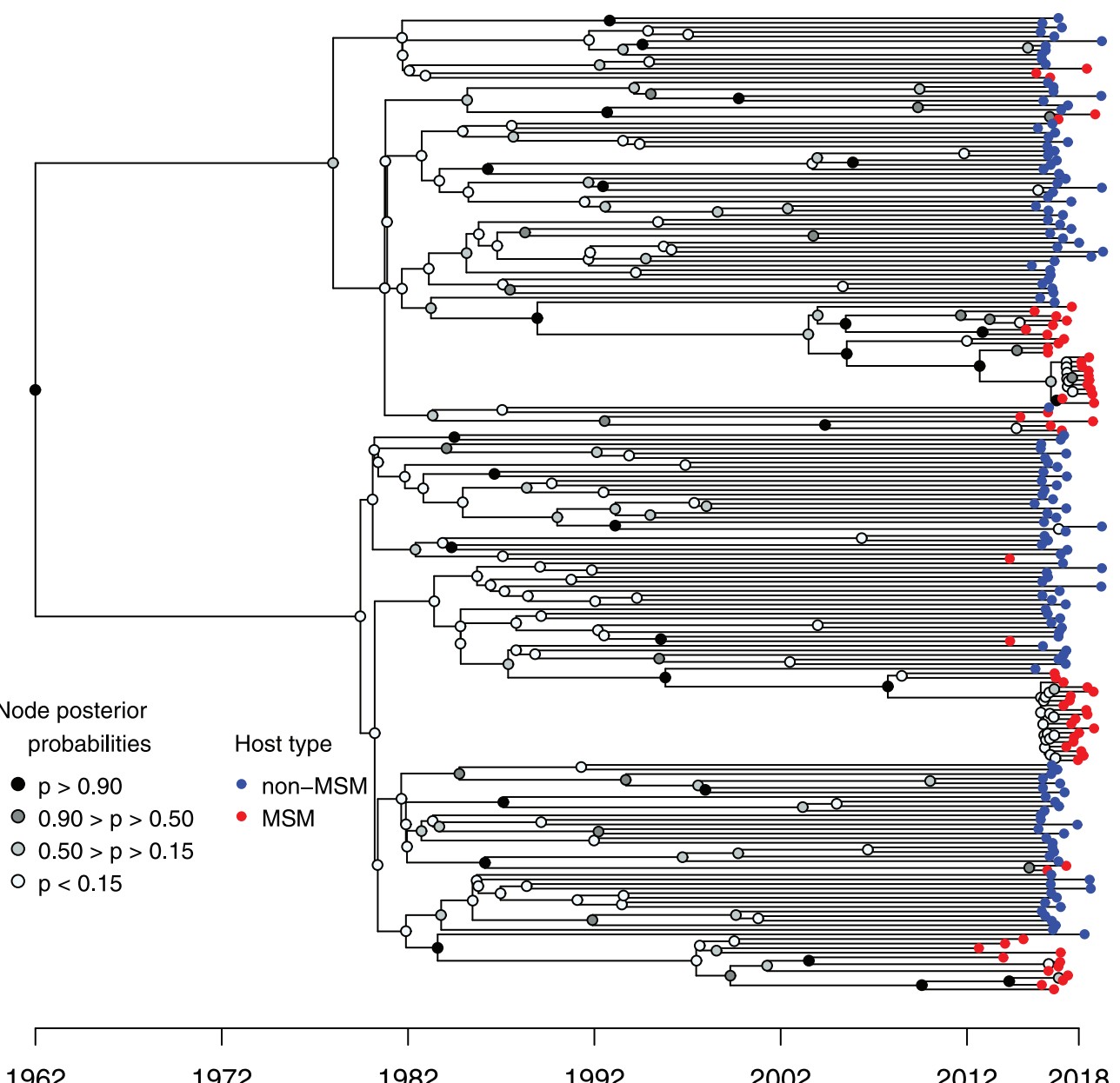

**Fig 1. Phylogeny of HCV infections in the area of Lyon (France).** Non-MSM hosts are in blue and MSM hosts are in red. Sampling events correspond to the end of black branches. The phylogeny was estimated using Bayesian inference (Beast2). The gray level of a node indicates its posterior probability. See the Material and methods for additional details.

The phylodynamics approach also allows us to infer the duration of the effective infectious period for each host type. Assuming that this parameter does not vary over time, we estimate it to be 3.85 years [1.09; 8.33] for non-MSM hosts (parameter $1/\gamma_1$) and 0.45 years [0.30; 0.77] for MSM hosts (parameter $1/\gamma_2$). We compute the ratio of $\gamma_2/\gamma_1$ and the 95% credibility interval does exclude 1.

Regarding effective reproduction numbers, i.e. the number of secondary infections caused by a given host over its infectious period, we estimate that of non-MSM hosts to have decreased from $R^{(1),t_1} = 1.96$ [1.45; 3.29] to $R^{(1),t_2} = 1.61$ [1.05; 2.08] after the introduction of

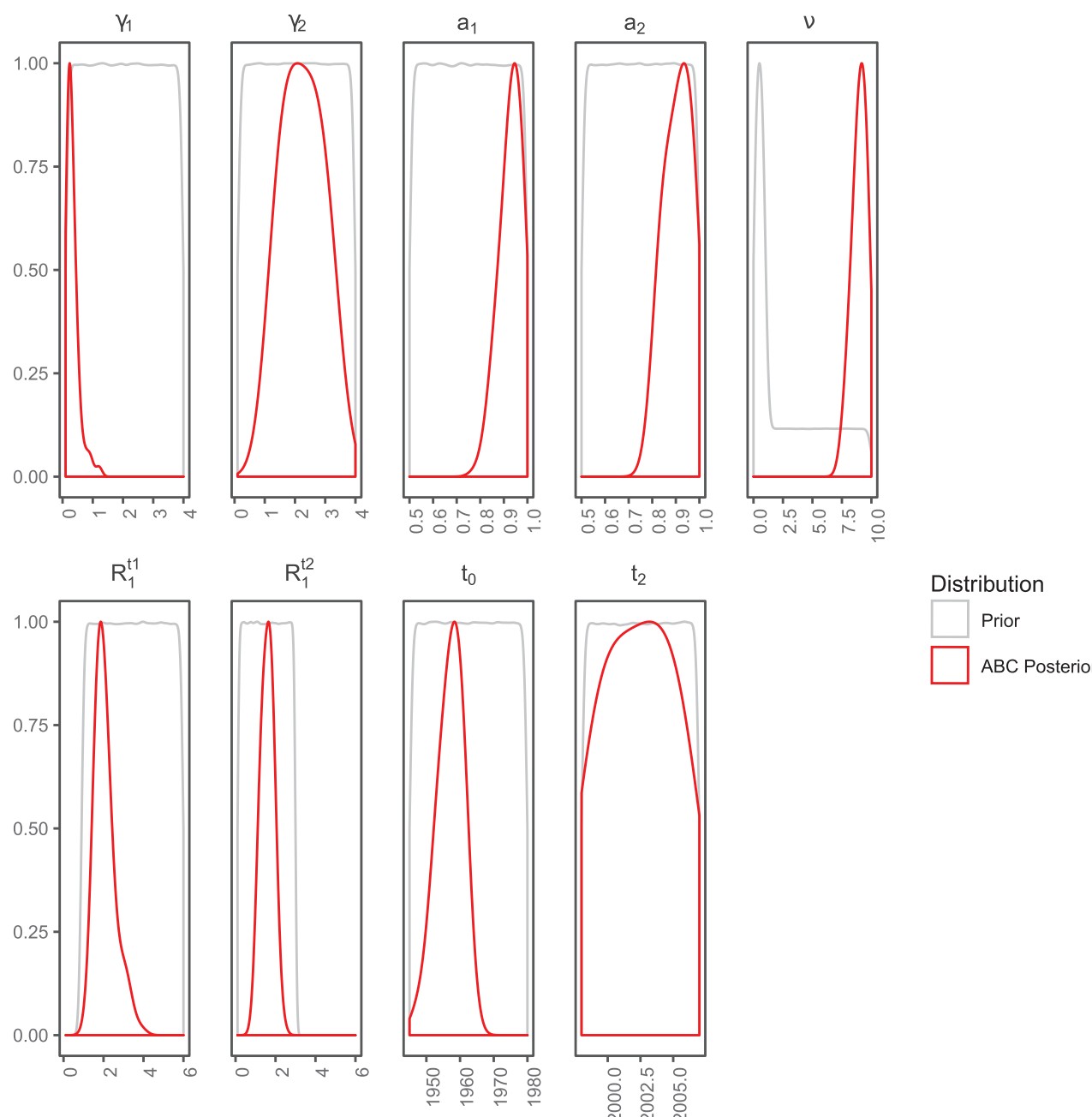

**Fig 2. Parameter prior and posterior distributions.** Prior distributions are in grey and posterior distributions inferred by ABC are in red. The thinner the posterior distribution width, the more precise the inference. Posterior distributions are truncated based on the prior distribution. The parameters $\gamma_1$ and $\gamma_2$ are the end of infectiousness rates for non-MSM and MSM hosts respectively. The parameters $a_1$ and $a_2$ are the assortativity levels between hosts types, for non-MSM and MSM hosts respectively. $\nu$ is the transmission rate differential between non-MSM and MSM hosts. The parameters $R_1^{t1}$ and $R_1^{t2}$ are respectively the reproduction numbers for the non-MSM hosts before and after the introduction of the third-generation HCV tests in 1997. The $t_0$ parameter is the origin of the epidemic in non-MSM hosts, and $t_2$ is the origin of the epidemic in MSM hosts.

the third-generation HCV test in 1997. The inference on the differential transmission parameter indicates that HCV transmission rate is $\nu = 9.0$ [7.7; 9.9] times greater from MSM hosts than from non-MSM hosts. By combining these results (see the Material and methods), we compute the effective reproduction number in MSM hosts and find $R^{(2),t_3} = 1.73$ [1.03; 4.32].

**Table 1. Table of inferred posterior distributions of parameters of the model.** Median values and 95% confidence interval of inferred posterior distributions of parameters of the model using the ABC approach.

|  | $\gamma_1$ | $\gamma_2$ | $a_1$ | $a_2$ | $v$ | $R_1^{t1}$ | $R_1^{t2}$ |
|---|---|---|---|---|---|---|---|
| median | 0.26 | 2.22 | 0.94 | 0.92 | 9.0 | 1.96 | 1.61 |
| 95% CI | [0.12; 0.92] | [1.29; 3.33] | [0.83; 1.0] | [0.81; 0.99] | [7.7; 9.9] | [1.45; 3.29] | [1.05; 2.08] |

We compute the ratio of the $R(t)$ of MSM hosts over the $R(t)$ of non-MSM hosts after 1997 and, the median value is 1.14 and the 95% credibility interval is [0.56; 3.25].

To better understand the differences between the two host types, we compute the epidemic doubling time ($t_D$), which is the time for an infected population to double in size. $t_D$ is computed for each type of host, assuming complete assortativity (see the Material and methods). We find that for the non-MSM hosts, before 1997 $t_D^{(1),t1} \approx 2.8$ years ([1.1; 5.0] years). After 1997, the pace decreases with a doubling time of $t_D^{(1),t2} \approx 4.4$ years ([2.0; 20.8] years). For the epidemics in the MSM hosts, we estimate that $t_D^{(2),t3} \approx 0.44$ years ([0.09; 8.84] years). When computing the ratio of the doubling times of non-MSM hosts after 1997 over the doubling times of the MSM hosts ($t_D^{(1),t2}/t_D^{(2),t3}$) to estimate the current difference we find that $t_D^{(1),t2}$ is 10 times higher than $t_D^{(2),t3}$ with a 95% credibility interval of [0.62; 149.99]. However, the 75% credibility interval does exclude 1 and is [3.39; 25.61]. Distributions for theses three doubling times are shown in S2 Fig.

To visualize the epidemiological dynamics, we simulated trajectories from the posterior distributions shown in Fig 3.

S3 Fig shows the correlations between parameters based on the posterior distributions. We mainly find that the $R(t)$ of non-MSM hosts after the introduction of the third generation of HCV detection tests (i.e. $R^{(1),t_2}$) is negatively correlated to $v$ and positively correlated to $\gamma_2$. In other words, the faster the epidemic spreads in non-MSM hosts ($R^{(1),t_2}$ is high), the slower the spread in MSM hosts (low $v$ or high $\gamma_2$) to explain the phylogeny (and vice versa). $R_0^{(1),t_2}$ is also slightly negatively correlated to $\gamma_1$, which most likely comes from the fact that for a given $R_0$, epidemics with a longer infection duration have a longer doubling time and therefore a weaker epidemiological impact. Overall, these correlations do not affect our main results, especially the pronounced difference in infection periods ($\gamma_1$ and $\gamma_2$).

To validate these results, we performed a goodness-of-fit test by simulating phylogenies using the resulting posterior distributions to determine whether these are similar to the target dataset (see the Material and methods). In Fig 4, we see that the target data in red, i.e. the projection of the observed summary statistics from the phylogeny shown in Fig 1, is contained in the envelope containing 90% of the simulations drawn from the posterior distributions. If we use the 95% HPD of the posterior but assume a uniform distribution instead of the true posterior distribution, we find that the target phylogeny is not contained in the envelope. These results confirm that the posterior distributions we infer are highly informative. In S4 Fig we show that for 77 summary statistics out of 101, the target value is in the 95% highest posterior

**Table 2. Table of computed posterior distributions of parameters of the model.** Median values and 95% confidence interval of posterior distributions of parameters computed from the inferred posterior distributions.

|  | $R_2^{t3}$ | $t_D^{(1),t1}$ | $t_D^{(1),t2}$ | $t_D^{(2),t3}$ |
|---|---|---|---|---|
| median | 1.73 | 2.80 | 4.40 | 0.44 |
| 95% CI | [1.03; 4.32] | [1.1; 5.0] | [2.0; 20.8] | [0.09; 8.84] |

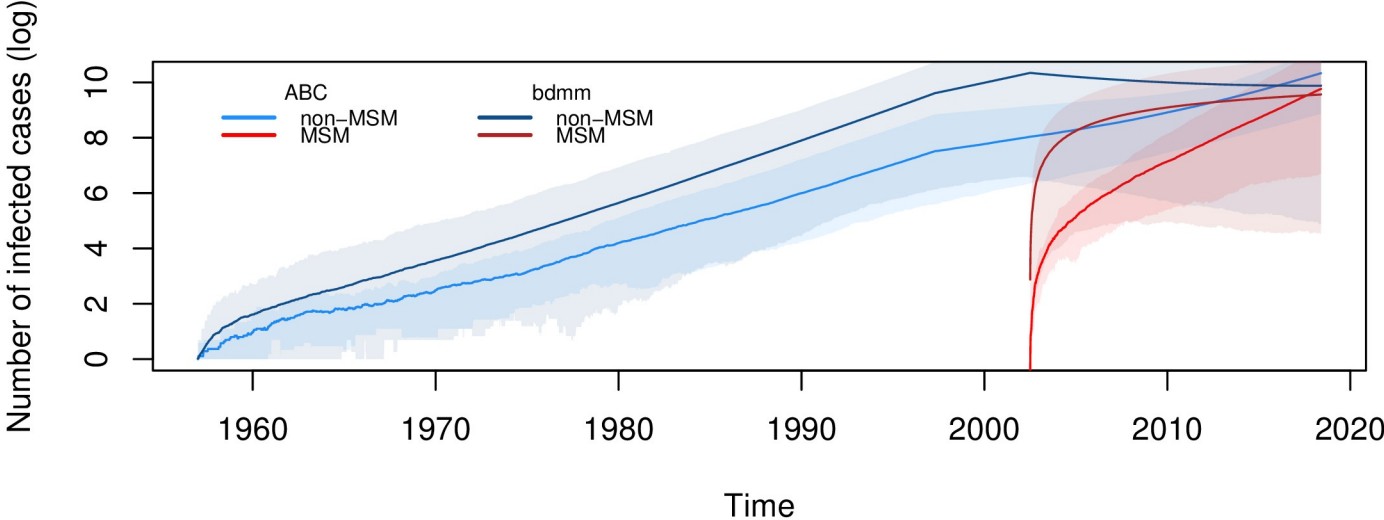

**Fig 3. Epidemiological trajectories inferred from the virus sequence data using ABC or Beast2.** The lines show the median natural logarithm values and the envelopes the 95% CI of trajectories corresponding to the number of infected cases through time, for each host type. Simulations were performed using the `TiPS` package from posterior distributions results of the ABC inference or the Beast2 `bdmm` inference.

distribution of summary statistics computed from the 10,000 simulated phylogenies from the posterior distribution used for the goodness-of-fit test.

To further explore the robustness of our inference method, we use simulated data to perform a 'leave one out' cross-validation (see the Material and methods). As shown in S5 Fig, the mean relative error made for each parameter inference is limited and comparable to what was found using a simpler SIR model [33]. One exception is for the MSM hosts' level of assortativity ($a_2$). This is likely due to the poor signal given the small size of the observed phylogeny.

A potential issue is that the sampling rate of MSM hosts may be higher than that of non-MSM hosts. To explore the effect of such sampling biases on the accuracy of our results, we sub-sampled the MSM hosts population by pruning the target phylogeny, i.e. randomly removing 50% of the MSM hosts' tips. In S6 Fig we show the posterior distributions estimated by our ABC method using the different pruned phylogenies. We find that although the confidence intervals are wider, the posterior distributions are all similar with the posterior distributions estimated using the target phylogeny. Finally, to evaluate the impact of phylogenetic reconstruction uncertainty, we analysed 100 additional trees from the Beast posterior distribution. In S7 Fig, we show that the estimates from our ABC method are qualitatively similar for all these trees.

## Discussion

Over the last years, the area of Lyon (France) witnessed an increase in HCV incidence both in HIV-positive and HIV-negative populations of men having sex with men (MSM) [17]. This increase appears to be driven by sexual transmission and echoes similar trends in Amsterdam [35] and Switzerland [36]. A quantitative analysis of the epidemic is necessary to optimise public health interventions. Unfortunately, this is challenging because the monitoring of the population at risk is limited and because classical tools in quantitative epidemiology, especially incidence time series, are poorly informative with such a heterogeneous population. To circumvent this problem, we used HCV sequence data, which we analysed using phylodynamics.

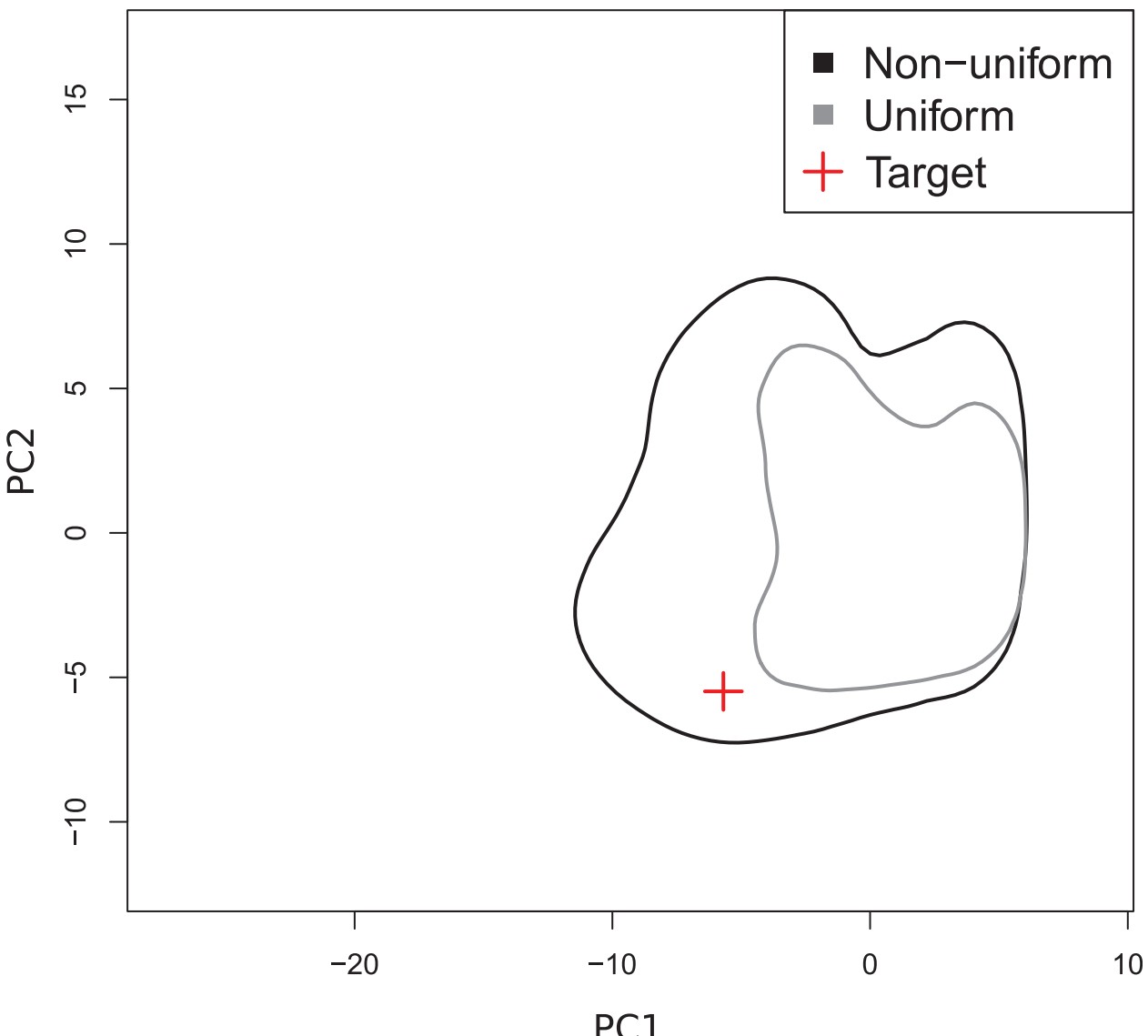

**Fig 4. Goodness-of-fit estimated using parameter bootstrap.** The graph displays envelopes containing 90% of the 10,000 simulations for each distribution. The envelope in black results from the posterior distribution, in grey, results from the uniform distribution drawn from the 95% HPD distribution. The target data is represented by a red cross. Axes units are based on the outcome of principal component analysis using the simulated summary statistics.

To account for host heterogeneity, we extended and validated an existing Approximate Bayesian Computation framework [33].

From a public health point of view, our results have two major implications. First, we find a strong degree of assortativity in both non-MSM and MSM host populations. The virus phylogeny does hint at this result (Fig 1) but the ABC approach allows us to quantify the pattern and to show that assortativity may be higher for non-MSM hosts. The second main result has to do with the striking difference in doubling times. Indeed, the current spread of the epidemics in MSM hosts appears to be five times more rapid than the spread in the non-MSM hosts in the early 1990s before the advent of the third generation tests in 1997, and ten times more rapid that the spread in the non-MSM hosts after 1997. That the duration of the effective infectious period in MSM hosts is in the same order of magnitude as the time until treatment suggests

that the majority of the transmission events may be occurring during the acute phase. This underlines the necessity to act rapidly upon detection, for instance by emphasising the importance of protection measures such as condom use and by initiating treatment even during the acute phase [37]. A better understanding of the underlying contact networks could provide additional information regarding the structure of the epidemics and, with that respect, next-generation sequence (NGS) data could be particularly informative [38–40].

The inferred phylogeny in Fig 1 suggests that the MSM epidemic is the result of multiple introductions. We inferred a phylogeny by adding sequences collected from MSM HCV infected patients from Amsterdam [41]. The phylogeny showed in S10 Fig shows that although the clades are monophyletic, the epidemics of MSM hosts from Lyon and from Amsterdam are potentially linked, with multiple 'migration' events between the two over the last decades. This phylogenetic analysis suggests that performing a phylogeographic study would be interesting to better understand the structure and history of French HCV epidemics, particularly since we know that there have been major epidemiological dynamics as shown by the circulation of different HCV genotypes [42].

Some potential limitations of the study are related to the sampling scheme, the assessment of the host type, and the transmission model. Regarding the sampling, the proportion of infected MSM host that is sampled is unknown but could be high. For the non-MSM hosts, we selected a representative subset of the patients detected in the area but this sampling is likely to be low. However, the effect of underestimating sampling for the MSM epidemics would be to underestimate its spread. Therefore, this would further increase our result that the MSM epidemic is spreading faster than the non-MSM epidemic. When running the analyses on different phylogenies with half of the MSM hosts sequences, we find results similar to those obtained with the whole phylogeny, suggesting that our ABC framework is partly robust to sampling biases. In general, implementing a more realistic sampling scheme in the model would be possible but it would require a more detailed model and more data to avoid identifiability issues. Regarding assigning hosts to one of the two types, this was performed by clinicians independently of the sequence data. The main criterion used was the infection stage (acute or chronic), which was complemented by other epidemiological criteria (history of intravenous drug use, blood transfusion, HIV status). Finally, the non-MSM and the MSM epidemics appear to be spreading on contact networks with different structures. However, such differences are beyond the level of details of the birth-death model we use here and would require a larger dataset for them to be inferred.

The inferred phylogeny (Fig 1) features two main clades from the MSM host epidemic with numerous branching events in recent time. These clades could correspond to epidemiological clusters that could bias the analysis, for instance leading to an overestimation of the MSM epidemic growth rate. To test whether these clades impact the analysis, we removed the sequences from these clades (i.e. represents 57% of the MSM sequences in the dataset), and performed an ABC inference analysis using the resulting phylogeny. The results are shown in S2 Table. Although the dataset is less informative, we find that the posterior distributions and the quantitative trends are similar with the results shown in the main text. In particular, we find the same difference between the duration of the infectious period of the MSM hosts and of the non-MSM hosts, along with a high level of assortativity. Furthermore, the inference of the differential transmission parameter ($v$) in this analysis indicates that HCV transmission rate is still greater from MSM hosts than from non-MSM hosts ($v \approx 7.0$).

To test whether the infection stage (acute vs. chronic) can explain the data better than the existence of two host types, we developed an alternative model without the two host types but where all infected hosts first go through an acute phase before recovering or progressing to the chronic phase. As for the model with two host types, we used three time intervals. S9 Fig

shows the diagram of the model as well as the corresponding equations. Interestingly, it was almost impossible to simulate phylogenies with this model. This is likely due to the intrinsic constrains on assortativity, which is absent in this alternative model since both acute and chronic infections generate new acute infections.

In our model, we assume that the duration of the infectious period for each host type ($1/\gamma_1$ and $1/\gamma_2$) remains constant through time. This assumption was motivated by the limited number of virus sequences and, hence, the limited inference power. Therefore, we decided to focus on variations in reproduction numbers. However, the implementation of non-pharmaceutical interventions after 1997 might have led to a decrease in the effective duration of the infectious period for the non-MSM hosts. To control for this bias, we reran the inference and estimated an additional parameter by allowing $1/\gamma_1$ to vary before and after 1997. The results we obtained show that the variation in the effective infection duration is limited but further data will be needed to investigate this trend in details.

Because we use a birth-death model, we *de facto* assume exponential growth in the number of infections. However, we also assume several time periods, which means that the epidemic can grow of decay exponentially. This is particularly important for the non-MSM epidemic, which appears to have originated in the 1960s. A potential limitation is that the reproduction number of the non-MSM epidemic is assumed to remain constant after 1997 whereas the MSM epidemic is assumed to originate after this date. An alternative, but less parsimonious approach, could have been to estimate the reproduction number for the non-MSM epidemic over 3 time periods instead of 2. To test the effect of this assumption, we performed the ABC analysis with an additional reproduction number for the non-MSM hosts in the most recent time period ($R_1^{t3}$). The results are shown in S3 Table and show that the reproduction number of the non-MSM hosts epidemic decreases monotonously. The other results regarding the rapid growth of the new MSM epidemic are unchanged.

To our knowledge, few attempts have been made in phylodynamics to tackle the issue of host population heterogeneity. In 2013, a study developed and used a maximum-likelihood method to a Latvian HIV-1 dataset to quantify the impact of the intravenous drug user epidemic on the heterosexual epidemic [43]. In 2018, a study used the structured coalescent model to investigate the importance of accounting for so-called 'superspreaders' in the recent Ebola epidemics in West Africa [44]. The same year, another study used the birth-death model to study the effect of drug resistance mutations on the $R_0$ of HIV strains [45]. Both of these are now implemented in Beast2 in the `bdmm` package [46]. In a methods comparison perspective, we ran this package with our data using a fixed phylogeny. Results in S1 Table and in Fig 3 show qualitatively similar results.

Overall, we show that our ABC approach, which we validated for simple SIR epidemiological models [33], can be applied to more elaborate models that current phylodynamics methods have difficulties to capture. Further increasing the level of details in the model may require to increase the number of simulations but also to introduce new summary statistics. Another promising perspective would be to combine sequence and incidence data. Although this could not be done here due to the limited sampling, such data integration can readily be done with regression-ABC.

## Material and methods

### Ethics statement

This study was conducted following French ethics regulations. All patients gave their written informed consent to allow the use of their personal clinical data. The study was approved by the Ethics Committee of Hospices Civils de Lyon.

## HCV sequence and epidemiological data

We included HCV molecular sequences of all MSM patients diagnosed with acute HCV genotype 1a infection at the Infectious Disease Department of the Hospices Civils de Lyon, France, and for whom NS5B sequencing was performed between January 2014 and December 2017 ($N = 68$). HCV genotype 1a isolated from $N = 145$ non-MSM, HIV-negative, male patients of similar age were analysed by NS5B sequencing at the same time for phylogenetic analysis. Host profiles have been established by field epidemiologists based on interviews and risk factors.

## HCV testing and sequencing

HCV RNA was detected and quantified using the Abbott RealTime HCV assay (Abbott Molecular, Rungis, France). The NS5B fragment of HCV was amplified between nucleotides 8256 and 8644 by RT-PCR as previously described and sequenced using the Sanger method. Electrophoresis and data collection were performed on a GenomeLab GeXP Genetic Analyzer (Beckman Coulter). Consensus sequences were assembled and analysed using the GenomeLab sequence analysis software. The genotype of each sample was determined by comparing its sequence with HCV reference sequences obtained from GenBank.

## Nucleotide accession numbers

All HCV NS5B sequences isolated in MSM and non-MSM patients reported in this study were submitted to the GenBank database. The list of Genbank accession numbers for all sequences is provided in S1 Appendix.

## Dated viral phylogeny

To infer the time-scaled viral phylogeny from the alignment we used a Bayesian Skyline model in `BEAST` v2.5.2 [47]. The general time-reversible (GTR) nucleotide substitution model was used with a strict clock rate fixed at $1.3 \cdot 10^{-3}$ substitutions/site/year based on data from Ref. [48] and a gamma distribution with four substitution rate categories. The MCMC was run for 100 million iterations and samples were saved every 100,000 iterations. We selected the maximum clade credibility using `TreeAnnotator BEAST2` package. The date of the last common ancestor was estimated to be 1961.95 with a 95% Highest Posterior Density (HPD) of [1941.846; 1975.516]. When performing the same inference without the MSM hosts, we found a similar estimate (1960) and the same 95% HPD of [1942; 1975], which we used as a prior distribution to estimate the origin of the non-MSM hosts $t_0$.

## Epidemiological model and simulations

We assume a Birth-Death model with two hosts types (S1 Fig) with non-MSM hosts (numbered 1) and MSM hosts (numbered 2). This model is described by the following system of ordinary differential equations (ODEs):

$$\frac{dI_1}{dt} = a_1 \beta I_1 + (1 - a_2) v \beta I_2 - \gamma_1 I_1 \tag{1a}$$

$$\frac{dI_2}{dt} = a_2 \beta v I_2 + (1 - a_1) \beta I_1 - \gamma_2 I_2 \tag{1b}$$

In the model, transmission events are possible within each type of hosts and between the two types of hosts at a transmission rate $\beta$. Parameter $v$ corresponds to the transmission rate differential between non-MSM and MSM hosts. Individuals can be 'removed' at a rate $\gamma_1$ from an

infectious compartment ($I_1$ or $I_2$) via infection clearance, host death or change in host behaviour (e.g. condom use). The assortativity between host types, which can be seen as the percentage of transmissions that occur with hosts from the same type, is captured by parameter $a_i$.

The effective reproduction number (denoted $R_t$) is the number of secondary cases caused by an infectious individual in a fully susceptible host population [34]. We seek to infer the $R_t$ from the non-MSM epidemic, denoted $R^{(1)}$ and defined by $R^{(1)} = \beta/\gamma_1$, as well as the $R(t)$ of the MSM epidemic, denoted $R^{(2)}$ and defined by $R^{(2)} = \nu\beta/\gamma_2 = \nu R^{(1)}\gamma_1/\gamma_2$.

The doubling time of an epidemic ($t_D$) corresponds to the time required for the number of infected hosts to double in size. It is usually estimated in the early stage of an epidemic when epidemic growth can be assumed to be exponential. To calculate it, we assume perfect assortativity ($a_1 = a_2 = 1$) and approximate the initial exponential growth rate by $\beta - \gamma_1$ for non-MSM hosts and $\nu\beta - \gamma_2$ for MSM hosts. Following [49], we obtain $t_D^{(1)} = \ln(2)/(\beta - \gamma_1)$ and $t_D^{(2)} = \ln(2)/(\nu\beta - \gamma_2)$.

We consider three time intervals. During the first interval $[t_0, t_1]$, $t_0$ being the year of the origin of the epidemic in the area of Lyon, we assume that only non-MSM hosts are present. The second interval $[t_1, t_2]$, begins in $t_1$ = 1997.3 with the introduction of the third generation HCV tests, which we assume to have affected $R^{(1)}$ through the decrease of the transmission rate $\beta$. Finally, the MSM hosts appear during the last interval $[t_2, t_3]$, where $t_2$, which we infer, is the date of origin of the second outbreak and is chosen in a uniform prior between $t_1$ and 2007. We assume the MSM hosts continuously emerge from the non-MSM host type. The final time ($t_3$) is set by the most recent sampling date in our dataset (2018.39). The prior distributions used are summarized in Table 3 and shown in Fig 2. Given the phylogeny structure suggesting a high degree of assortativity, we assume the assortativity parameters, $a_1$ and $a_2$, to be higher than 50%. For the prior distribution of parameter $\nu$, we combined a uniform distribution from 0 to 1 with a uniform distribution from 1 to 10. This was done to ensure that the probability to have $\nu < 1$ is equal to the probability to have $\nu > 1$. Given the number of virus sequences and the potential limitations in inference power, we fixed the upper limit of the prior of $\nu$ to 10.

To simulate phylogenies, we use our `TiPS` simulator [50] implemented in `R` via the `Rcpp` package. This is done in a two-step procedure. First, epidemiological trajectories are simulated using the compartmental model in Eq 1 and Gillespie's stochastic event-driven simulation algorithm [51]. The number of individuals in each compartment and the reactions occurring through the simulations of trajectories, such as recovery or transmission events, are recorded. Using the target phylogeny, we know when sampling events occur. For each simulation, each sampling date is randomly associated to a host compartment using the observed fraction of each infection type (here 68% of the dates associated with non-MSM hosts type and 32% with MSM hosts). Once the sampling dates are added to the trajectories, we move to the second step, which involves simulating the phylogeny. This step starts from the last sampling date and follows the epidemiological trajectory through a coalescent process, that is backwards-in-time. Each backward step in the trajectory can induce a tree modification given a probability and

**Table 3. Prior distributions for the birth-death model parameters over the three time intervals.** $t_0$ is the date of origin of the epidemics in the studied area, $t_1$ is the date of introduction of $3^{rd}$ generation HCV tests, $t_2$ is the date of emergence of the epidemic in MSM hosts and $t_3$ is the time of the most recent sampled sequence.

| Interval | $\gamma_i$ | $\nu$ | $R^{(1)}$ | $a_i$ |
|---|---|---|---|---|
| $[t_0, t_1]$ | Unif(0.1, 4) | 0 | Unif(0.9, 6) | Unif(0.5, 1) |
| $[t_1, t_2]$ | | | Unif(0.1, 3) | |
| $[t_2, t_3]$ | | Unif(0, 1) & Unif(1, 10) | | |

the population size: a sampling event leads to a labelled leaf in the phylogeny, a transmission event can lead to the coalescence of two sampled lineages or to no modification of the phylogeny (if one of the lineages is not sampled).

We implicitly assume that the sampling rate is low, which is consistent with the limited number of sequences in the dataset. We also assume that the virus can still be transmitted after sampling.

We simulate 60, 000 phylogenies from known parameter sets drawn in the prior distributions shown in Table 3. These are used to perform the rejection step and build the regression model in the Approximate Bayesian Computation (ABC) inference.

## ABC inference

**Summary statistics.**    Phylogenies are rich objects and to compare them we break them into summary statistics. These are chosen to capture the epidemiological information of interest. In particular, following an earlier study, we use summary statistics from branch lengths, tree topology, and lineage-through-time (LTT) [33], and summary statistics based on the Laplacian spectrum using the `spectR` function of the `RPANDA` R package [52].

We also compute new summary statistics to extract information regarding the heterogeneity of the population, the assortativity, and the difference between the two $R(t)$. To do so, we annotate each internal node by associating it with a probability to be in a particular state (here the host type, non-MSM or MSM). We assume that this probability is given by the ratio

$$P(Y) = \frac{\text{number of descendent leaves labelled } Y}{\text{number of descendent leaves}} \tag{2}$$

where $Y$ is a state (or host type). Each node is therefore annotated with $n$ ratios, $n$ being the number of possible states. Since in our case $n = 2$, we only follow one of the labels and use the mean and the variance of the distribution of the ratios (one for each node) as summary statistics.

In a phylogeny, cherries are pairs of leaves that are adjacent to a common ancestor. There are $n(n + 1)/2$ categories of cherries. Here, we compute the proportion of homogeneous cherries for each label and the proportion of heterogeneous cherries. We also consider pitchforks, which we define as a cherry and a leaf adjacent to a common ancestor, and introduce three categories: homogeneous pitchforks, pitchforks whose cherries are homogeneous for a label and whose leaf is labelled with another trait, and pitchforks whose cherries are heterogeneous.

The Lineage-Through-Time (LTT) plot displays the number of lineages of a phylogeny over time. In this plot, the number of lineages is incremented by one every time there is a new branch in the phylogeny and is decreased by one every time there is a new leaf in the phylogeny. We use the ratios defined for each internal node to build an LTT plot for each label type, which we refer to as 'LTT label plot'. After each branching event in phylogeny, we increment the number of lineages by the value of the ratio of the internal node for the given label. This number of lineages is decreased by one every time there is a leaf in the phylogeny. In the end, we obtain $n = 2$ LTT label plots.

Finally, for each label, we compute some of our branch lengths summary statistics on homogeneous clades and heterogeneous clades present in the phylogeny. Homogeneous clades are defined by their root having a ratio of 1 for one type of label and their size being greater than $N_{\min}$. For heterogeneous clades, we keep the size criterion and impose that the ratio is smaller than 1 but greater than a threshold $\epsilon$. After preliminary analyses, we set $N_{\min} = 4$ leaves and $\epsilon = 0.7$. We then obtain a set of homogeneous clades and a set of heterogeneous clades, the branch lengths of which we pool into two sets to compute the summary statistics of

heterogeneous and homogeneous clades. Note that we always select the largest clade, for both homogeneous and heterogeneous cases, to avoid redundancy.

**Regression-ABC.**   We first measure multicollinearity between summary statistics using variance inflation factors (VIF). Each summary statistic is kept if its VIF value is lower than 10. This stepwise VIF test leads to the selection of 101 summary statistics out of 330.

We then use the `abc` function from the `abc R` package [53] to infer posterior distributions generated using only the rejection step. Finally, we perform linear adjustment using an elastic net regression.

The `abc` function performs a classical one-step rejection algorithm [32] using a tolerance parameter $P_\delta$, which represents a percentile of the simulations that are close to the target. To compute the distance between a simulation and the target, we use the Euclidian distance between normalized simulated vectors of summary statistics and the normalized target vector.

Before linear adjustment, the `abc` function performs smooth weighting using an Epanechnikov kernel [32]. Then, using the `glmnet` package in `R`, we implement an elastic-net (EN) adjustment, which balances the Ridge and the LASSO regression penalties [54]. Since the EN performs a linear regression, it is not subject to the risk of over-fitting that may occur for non-linear regressions (e.g. when using neural networks, support vector machines or random forests).

In the end, we obtain posterior distributions for $t_0$, $t_2$, $a_1$, $a_2$, $v$, $\gamma_1$, $\gamma_2$, $R^{(1),t_1}$ and $R^{(1),t_2}$ using our ABC-EN regression model with $P_\delta = 0.05$.

**Parametric bootstrap and cross-validation.**   Our goodness-of-fit validation consists in simulating 10, 000 additional phylogenies from parameter sets drawn in posterior distributions. We then compute summary statistics and perform a goodness of fit using the `gfitpca` function from the `abc R` package [53]. The function performs principal component analysis (PCA) using the simulated summary statistics. It displays envelopes containing a given percentage, here 90%, of the simulations. The projection of the observed summary statistics is displayed to check if they are contained or not in the envelopes. If the posterior distribution is informative, we expect the target data to be contained in the envelope. This analysis was performed either on the posterior distribution, or on a uniform distribution based on the 95% HPD posterior distribution of each parameter, the latter being less informative.

To assess the robustness of our ABC-EN method to infer epidemiological parameters of our BD model, we also perform a 'leave-one-out' cross-validation as in [33]. This consists in inferring posterior distributions of the parameters from one simulated phylogeny, assumed to be the target phylogeny, using the ABC-EN method with the remaining 59, 999 simulated phylogenies. We run the cross-validation 100 times with 100 different target phylogenies. We consider three parameter distributions $\theta$: the prior distribution, the prior distribution reduced by the feasibility of the simulations and the ABC inferred posterior distribution. For each of these parameter distributions, we measure the median and compute, for each simulation scenario, the mean relative error (MRE) such as:

$$MRE = \frac{1}{100} \sum_{i=1}^{100} \left| \frac{\theta_i}{\Theta} - 1 \right| \tag{3}$$

where $\Theta$ is the true value.

## Supporting information

**S1 Fig. Diagram of the birth-death model with host heterogeneity.**
(PDF)

**S2 Fig. Densities of the inferred doubling times.** The density of the doubling time for the non-MSM hosts before 1997 ($t_D^{(2),t1}$) is in blue dashed line, and after 1997 ($t_D^{(1),t2}$) in blue solid line. The density of the doubling time for the MSM hosts ($t_D^{(2),t3}$) is in red.
(PDF)

**S3 Fig. Correlation heat map between the posterior distributions for the model parameters.** The intensity of the colour is proportional to the correlation coefficients.
(PDF)

**S4 Fig. Distributions of selected summary statistics.** The dots represent the median and the horizontal lines represent the 95% HPD. Red distributions correspond to the summary statistics computed from the 10,000 phylogenies simulated from the posterior distribution. Black dots represent the values of selected summary statistics computed from the target phylogeny. Summary statistics are represented by group.
(PDF)

**S5 Fig. Cross-validation results.** Each column corresponds to one of the inferred parameters. The first line shows the prior distribution. The second line shows the distribution of values for which a phylogeny could be simulated. The third line shows the inference after the ABC. For the rejection step of the ABC, the tolerance level was set to $P_\delta = 0.05$. The rectangles show the mean relative errors and their standard errors computed for 100 target sets with known values (see the Material and methods).
(PDF)

**S6 Fig. Posterior distributions estimated from different phylogenies inferred using half of the MSM hosts' sequences.** The first line represents the prior (in grey), the last line the full target tree (in red), and all the intermediate lines phylogenies where half of the MSM hosts' sequences were removed at random.
(PDF)

**S7 Fig. Variation in posterior distribution estimated from different inferred phylogenies.** The dots represent the median and the horizontal lines represent the 95% highest posterior density (HPD) of each distribution. Grey distributions correspond to the prior, orange distributions correspond to the different posterior distributions computed from 100 phylogenies drawn at random in the posterior distribution of trees inferred by Beast2 and red distributions correspond to the ABC-EN posterior distributions.
(PDF)

**S8 Fig. Density distributions of the $t_{MRCA}$ for the observed Beast2 phylogeny (in black) and for the 100 phylogenies drawn at random in the posterior distributions of trees inferred by Beast2 (in red).**
(PDF)

**S9 Fig. Diagram of the alternative model where all infected hosts first go through an acute phase ($A_i$) before recovering or progressing to the chronic phase ($C_i$).** $\omega$ is the proportion of infections that clear before becoming chronic, $\sigma$ is the rate at which acute infections become chronic, and other parameters are identical to those in the main text. The equations governing the dynamics of the system can be written as $\frac{dA_i}{dt} = a_i\beta_i(A_i + C_i) + (1 - a_j)\beta_j(A_j + C_j) - \sigma A_i$ and $\frac{dC_i}{dt} = \sigma(1 - \omega)A_i - \gamma_i C_i$ with $i \neq j$, $\beta_1 = \beta$ and $\beta_2 = \nu\beta$.
(PDF)

**S10 Fig. Phylogeny of HCV infections in Europe using sequences from the area of Lyon and sequences from Amsterdam.** Non-MSM hosts from Lyon are in blue and MSM hosts from Lyon are in red. MSM hosts' sequences from Amsterdam are in green. Sampling events correspond to the end of black branches. The phylogeny was estimated using Bayesian inference (Beast2).
(PDF)

**S1 Table. Median values and 95% confidence interval of the posterior distributions of the inferred parameters using the bdmm BEAST2 package.** The parameters $R_1^{t1}$, $R_1^{t2}$ and $R_1^{t3}$ are the reproduction numbers for the non-MSM hosts during the first, second and last temporal intervals respectively. The parameter $R_2^{t3}$ is the reproduction number for the MSM hosts epidemic. $\gamma_1$ and $\gamma_2$ are the end of infectiousness rates of non-MSM and MSM hosts respectively. $t_2$ corresponds to the date of the emergence of the MSM hosts epidemic.
(PDF)

**S2 Table. Table presenting the mean, median values and 95% confidence interval of the inferred posterior distributions of parameters and computed posterior distributions of parameters of the model using a phylogeny with MSM hosts' sequences corresponding to the two main clades being removed.**
(PDF)

**S3 Table. Table presenting the mean, median values and 95% confidence interval of the inferred posterior distributions of parameters and computed posterior distributions of parameters of the model.** In this analysis, the reproduction number of the non-MSM hosts in the most recent time period ($R_1^{t3}$) is also inferred.
(PDF)

**S1 Appendix. HCV sequence accession numbers.**
(PDF)

## Acknowledgments

We thank Jūlija Pečerska for her help with Beast2. GD and SA acknowledge further support from the CNRS, the IRD and the itrop HPC (South Green Platform) at IRD montpellier, which provided HPC resources that contributed to the results reported here (https://bioinfo.ird.fr/).

## Author Contributions

**Conceptualization:** Gonché Danesh, Victor Virlogeux, Christophe Ramière, Caroline Charre, Laurent Cotte, Samuel Alizon.

**Data curation:** Victor Virlogeux, Christophe Ramière, Caroline Charre, Laurent Cotte.

**Formal analysis:** Gonché Danesh.

**Funding acquisition:** Samuel Alizon.

**Investigation:** Gonché Danesh.

**Methodology:** Gonché Danesh, Victor Virlogeux, Laurent Cotte, Samuel Alizon.

**Project administration:** Samuel Alizon.

**Resources:** Gonché Danesh, Victor Virlogeux, Christophe Ramière, Caroline Charre, Laurent Cotte, Samuel Alizon.

**Supervision:** Laurent Cotte, Samuel Alizon.

**Validation:** Gonché Danesh, Laurent Cotte, Samuel Alizon.

**Writing – original draft:** Gonché Danesh, Samuel Alizon.

**Writing – review & editing:** Gonché Danesh, Victor Virlogeux, Christophe Ramière, Caroline Charre, Laurent Cotte, Samuel Alizon.

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
