## [Decision Letter · Decision Letter 0]

17 Feb 2021

Dear Mrs. Danesh,

Thank you very much for submitting your manuscript "Quantifying transmission dynamics of acute hepatitis C virus infections in a heterogeneous population using sequence data" for consideration at PLOS Pathogens. As with all papers reviewed by the journal, your manuscript was reviewed by members of the editorial board and by several independent reviewers. In light of the reviews (below this email), we would like to invite the resubmission of a significantly-revised version that takes into account the reviewers' comments.

We cannot make any decision about publication until we have seen the revised manuscript and your response to the reviewers' comments. Your revised manuscript is also likely to be sent to reviewers for further evaluation.

Sincerely,

Nels C. Elde, Ph.D.

Associate Editor

PLOS Pathogens

Marco Vignuzzi

Section Editor

PLOS Pathogens

Kasturi Haldar

Editor-in-Chief

PLOS Pathogens

orcid.org/0000-0001-5065-158X

Michael Malim

Editor-in-Chief

PLOS Pathogens

orcid.org/0000-0002-7699-2064

Reviewer's Responses to Questions

**Part I - Summary**

Reviewer #1: This is a very interesting paper that describes a novel phylodynamic approach. The authors do not explicitly combine an evolutionary and epidemiological model into one approach, in the manner of more popular coalescent and birth-death phylodynamic models (e.g. the work of Erik Volz, Tanja Stadler, David Rasmussen, etc.). Rather, the authors use Approximate Bayesian Computation to evaluate sets of epidemiological parameters using a given phylogeny as the “target” phylogeny (the empirical, known, phylogeny is considered fixed before the ABC is started). This really is quite a fascinating study, and while I do not have the full expertise to comment exhaustively on all of the statistical approaches included, I believe it will be an important addition to the literature. It is true that genomic epidemiology has hesitated to use phylogenies as calibration tools (Art Poon’s work with HIV notwithstanding, and Emma Saulnier et al’s work from 2017), but this might be an important tool in the near future.

Reviewer #2: In this manuscript, Danesh et al. apply Approximate Bayesian Computation to time collaborated Hepatitis-C (HCV) phylogenies to estimate key epidemiological parameters of the recent HCV epidemic in Lyon, France. HCV is currently spreading in a heterologous population of “classical” hosts (i.e. those infected as the result of intravenous drug use or blood transfusion) and “new” hosts, typically MSM. Viral phylogenetics disentangles these co-occurring epidemics and shows that they are driven by two separate processes in separate populations. The authors then use ABC to estimate the parameters of a more complex compartmental model than is feasible using classical phylodynamic approaches. The approach is well-suited to the problem of dissecting and characterizing the recent epidemic, and the findings provide important insights into a growing public health concern. The fact that the “new” and “classical” epidemics are distinct events that rarely mix is an important and robust finding. However, there are caveats to the specific claims made regarding the parameter values that detract from an otherwise sound study. In particular, the placement of the phylogenetic tree in the context of global HCV circulation and the impact of the time constraints imposed on the model should be explored.

Reviewer #3: The authors use phylodynamic methods to shed light on the HCV epidemic oin Lyon, France. They identify two host types, classical and new ones, the latter of which appears to have a drastically faster epidemic doubling time, calling for a quick response by public health authorities. This study is important and well within the scope of this journal.

**Part II – Major Issues: Key Experiments Required for Acceptance**

Reviewer #1: No major issues.

Reviewer #2: 1) The analysis relies on a phylodynamic reconstruction of hosts sampled only in the recent past and assumes a single exponentially growing epidemic in Lyon since the late 1960s. What is known about HCV import into Lyon during the decades covered by the phylogeny? Does the early growth indicated by the phylogeny correspond to endemic growth in Lyon, or is it the result of a combination of endemic growth and foreign epidemics that were imported into Lyon? Earlier samples and samples from different geographic regions should be included to verify the dataset represents a single epidemic in Lyon that was seeded in the 1960s. (These context-samples could be pruned out before fitting the ABC model).

Related to the point above, the recent, “new” host epidemic appears to be driven by two heavily sampled clades, which have a large number of branching events in the very recent past. The authors account for differences in sampling proportion between demes; however, they do not account for biased sampling within demes. If these clades are known or probable epidemiological clusters then including them in the analysis would lead to an artificial increase in the growth rate of the “new” epidemic. Is anything known about these clusters? It would be interesting to quantify the impact these clades have on the growth estimates by removing them from the tree and estimating their growth separately.

2) The authors compare the doubling time of the “new” epidemic with that of the “classical” epidemic after the advent of rapid tests in 1997. There are several issues with this comparison. The method used here assumes an exponentially growing epidemic; however, the “classical” epidemic is decades old at this point. The authors should discuss the impact of these assumptions on their estimates. Also, one of the parameters in this comparison, gamma, is assumed to be constant throughout the epidemic. This seems valid for the “new” epidemic which is young and growing, but not for the “classic” one which spans 60 years and has been impacted by non-pharmaceutical interventions. As the main conclusions stem from a difference in the gamma parameters any bias in these estimates should carefully be considered.

3) The manuscript reports that the transmissibility of the “new” epidemic (nu) is ~10 times that of the classical epidemic. This estimate appears to be at the limit imposed by the prior. Is there any reason to limit the prior to a range of 0,10? If not, a more diffuse prior should be used.

The estimate also relies on determining the origin of the “new” epidemic (t2) (which the authors state can not be estimated accurately) and compares parameters in this time frame to those driving the “classical” epidemic in other periods. The methods are unclear, are the classical parameters also estimated in the final time range [t2,t3]? If not, why not. The authors should discuss what this origin represents. The tree suggests that the “new” epidemic is the result of multiple introductions into the “new” population, not a single origin event. Judging by the tree in Figure 1, some of these introductions could date back to the late 1970s. However, by setting nu to 0 until the early 2000s, the “new” epidemic is fixed at no growth until the recent past. What is the impact of estimating the growth of the “new” epidemic over the entire period shown by the tree?

Reviewer #3: 1) a) Please rename "new" and "classic" hosts - otherwise this will be outdated soon.

b) The group assignment seems difficult, since "chemsex" has recently been associated with injecting drug use as well. Please discuss this more thoroughly.

2) Could the estimated difference in doubling times be biased by the fact that the "new" host epidemic started very recently? Please test this with simulations in which "classical" hosts have a short doubling time and "new hosts" have a long doubling time (but with a higher sampling rate in new hosts than classical ones, as the authors say that the "sampling rate of ’new’ hosts may be higher than that of 144 ’classical’ hosts" l.144)

3) (related to 1 & 2) The authors state that the type assignment was mainly done by infection stage (acute vs chronic) - please perform a sensitivity test in which the acute infections in the third (most recent) time interval are denoted as "new" hosts, to test if this drives the difference in epidemic doubling time. the sensitivity test the authors did is not enough, because it would also assign older acute infections to the new host type (which does not seem to reflect the assignment in the real data)

4) The estimated recovery rates may be biased by the lack of a sampling process in birth-death model? If sampling is much faster in new vs classical hosts, this will lead to the recovery rate being estimated as faster as well, and with that reduce the epidemic doubling time. If the ABC implementation doesn't allow for that, this could be tested in bdmm (with the target tree fixed)

5) Do the authors have any data on further co-infections e.g. with syphilis - as this increases the risk of transmission? How common are such co-infections in the study population? Please add to the discussion.

**Part III – Minor Issues: Editorial and Data Presentation Modifications**

Reviewer #1: 1. Page 1 in the Background, the authors describe incidence patterns across time and across populations, but don’t state the actual incidence estimates, or case counts. These would be helpful for context.

2. For Figure 2, in the legend, can you add the definitions of the variables?

3. Re: “The phylodynamics approach also allows us to infer the duration of the infectious

period for each host type.”

and “3.85 years [1.09;8.33] for ‘classical’ hosts (parameter 1/γ1) and 0.45 years [0.30;0.77] for ‘new’ hosts.”

The results for the duration of the infectious period seem odd, if not in results then in simple definition. I would have thought that the duration of the infectious period is generally a static parameter, similar across populations and time, that has more to do with intrahost virological and immunological processes than with epidemiological or network processes. But the authors' inference of such large differences in the infectious period between the 'classical' and 'new' hosts implies that something virological is underlying this pattern. Maybe I am simply misunderstanding their model and terminology, but this seems to need a bit more discussion. The SIR model includes this parameter as γ1 and γ2, and defines it as the rate of removal from the infectious compartment (I1 or I2) via infection clearance, host death, or change in host behaviour (e.g. condom use). This definition makes sense, and is consistent with standard SIR models. So maybe I am just quibbling about the terminology.

4. I read with interest the description of the authors' attempt to use BEAST2 (and jointly estimate the phylogeny and epidemiological parameters) for their dataset. And the sentence, "We were unable to conclude anything from 215 this analysis." is really something. At the risk of making the paper longer, it would be informative to hear what aspects of the HCV data that the authors thought was at fault for the inability of these methods to help. Particularly the structured coalescent approach.

Reviewer #2: Abstract: “new” and “classical” hosts should be defined before being used.

It would be very useful to show the inferred trajectories of the number of cases in both populations overtime. As the authors state, these metrics are difficult to interpret in a heterogeneous population without phylogenetic approaches. It would be nice to see how the presented analysis augments classical approaches.

It would be nice to include a table of the key parameter estimates (gamma, beta, nu, a) and their statistics (R0 and tD) for each population and period.

Related to (3) above, what are the initial, infected population, sizes used in the model, and do they account for the multiple introductions into the “new” host population before the estimated origin of that epidemic (t2)?

The tree in Figure 1 forms the basis of the study and is generated from ~400 basepair region of the genome. There is likely a large amount of uncertainty in the topology and node heights. Posterior probabilities and / node heigh bars should be shown in some way to indicate uncertainly in the data.

Supplemental Figure 3: It would be nice to see the beta parameters directly included in the correlation matrix.

The set of 100 sampled trees is a nice addition. Is it feasible to estimate the one distribution of parameter values across all these trees as opposed to estimating the parameters for each one separately?

Supplemental Figure 8: Is there any explanation for why the distribution of the tMCRA dates of the 100 randomly sampled trees does not match that of the posterior, other than unlucky sampling?

Line 174) “new” should be “classical”

Figure 2 legend: The thinner the posterior distributions represent more precise estimates but not necessarily more accurate ones.

Reviewer #3: - Please clarify if all 'classical' hosts from Lyon, too?

- l.258 "strict clock rate fixed at 1.3 · 10−3 based on data from Ref. [44] "

- l 279: doubling time estimation assumes we're still in exponential phase, which is true for new hosts but not for classical ones - please add as caveat to discussion

- Table 1: Please add which prior is used on t2.

- How were the priors for R1 chosen (interval 1: Unif(0.9, 6); interval 2: Unif(0.1, 3))?

- S9_Fig is missing arrows from C1>A1 and C2>A1 indicating transmission from chronic to acute individuals within the same host compartment

- ll. 119-120 "if the epidemic spreads rapidly in ‘classical’ hosts, it requires a slower spread in ‘new’ hosts to explain the phylogeny" - and vice versa (which seems to be the case here)

- l. 123: "LONGER doubling time" would be more precise than "lower doubling time"

- ll.188-189: "which is already faster than the classical epidemics" - please clarify what is meant here

- ll 209ff: Stadler & Bonhoeffer did this already in 2013, doi: 10.1098/rstb.2012.0198 using an ML approach on fixed trees

- l 215: the problems described here may be overcome by using bdmm on the fixed target tree, which would be good to have for comparison. there is no doubt that the approach presented here is useful, but the authors seem to imply here that there are no good alternatives, which is not true. please rephrase.

PLOS authors have the option to publish the peer review history of their article (what does this mean?). If published, this will include your full peer review and any attached files.

Reviewer #1: No

Reviewer #2: No

Reviewer #3: No
---

## [Editor Report · Decision Letter 1]

25 Aug 2021

Dear Mrs. Danesh,

We are pleased to inform you that your manuscript 'Quantifying transmission dynamics of acute hepatitis C virus infections in a heterogeneous population using sequence data' has been provisionally accepted for publication in PLOS Pathogens.

Best regards,

Nels C. Elde, Ph.D.

Associate Editor

PLOS Pathogens

Marco Vignuzzi

Section Editor

PLOS Pathogens

Kasturi Haldar

Editor-in-Chief

PLOS Pathogens

orcid.org/0000-0001-5065-158X

Michael Malim

Editor-in-Chief

PLOS Pathogens

orcid.org/0000-0002-7699-2064
---

## [Editor Report · Acceptance letter]

7 Sep 2021

Dear Mrs. Danesh,

We are delighted to inform you that your manuscript, "Quantifying transmission dynamics of acute hepatitis C virus infections in a heterogeneous population using sequence data," has been formally accepted for publication in PLOS Pathogens.

Best regards,

Kasturi Haldar

Editor-in-Chief

PLOS Pathogens

orcid.org/0000-0001-5065-158X

Michael Malim

Editor-in-Chief

PLOS Pathogens

orcid.org/0000-0002-7699-2064